# The Impact of Edema on MRI Radiomics for the Prediction of Lung Metastasis in Soft Tissue Sarcoma

**DOI:** 10.3390/diagnostics13193134

**Published:** 2023-10-05

**Authors:** Roberto Casale, Riccardo De Angelis, Nicolas Coquelet, Ayoub Mokhtari, Maria Antonietta Bali

**Affiliations:** Institut Jules Bordet Hôpital Universitaire de Bruxelles, Université Libre de Bruxelles, 1070 Brussels, Belgium; roberto.casale@hubruxelles.be (R.C.); riccardo.deangelis@hubruxelles.be (R.D.A.); nicolas.coquelet@hubruxelles.be (N.C.); maria.bali@hubruxelles.be (M.A.B.)

**Keywords:** radiomics, magnetic resonance imaging (MRI), soft tissue sarcoma, lung metastasis, edema

## Abstract

**Highlights:**

**What are the main findings?**
Both the model utilizing edema-related features and the model utilizing mass-related features demonstrated promising results in predicting the occurrence of lung metastases, with similar performances.

**What is the implication of the main finding?**
The findings suggest that the analysis of radiomic features extracted exclusively from edema can offer valuable insights into the prediction of lung metastases.

**Abstract:**

Introduction: This study aimed to evaluate whether radiomic features extracted solely from the edema of soft tissue sarcomas (STS) could predict the occurrence of lung metastasis in comparison with features extracted solely from the tumoral mass. Materials and Methods: We retrospectively analyzed magnetic resonance imaging (MRI) scans of 32 STSs, including 14 with lung metastasis and 18 without. A segmentation of the tumor mass and edema was assessed for each MRI examination. A total of 107 radiomic features were extracted for each mass segmentation and 107 radiomic features for each edema segmentation. A two-step feature selection process was applied. Two predictive features for the development of lung metastasis were selected from the mass-related features, as well as two predictive features from the edema-related features. Two Random Forest models were created based on these selected features; 100 random subsampling runs were performed. Key performance metrics, including accuracy and area under the ROC curve (AUC), were calculated, and the resulting accuracies were compared. Results: The model based on mass-related features achieved a median accuracy of 0.83 and a median AUC of 0.88, while the model based on edema-related features achieved a median accuracy of 0.75 and a median AUC of 0.79. A statistical analysis comparing the accuracies of the two models revealed no significant difference. Conclusion: Both models showed promise in predicting the occurrence of lung metastasis in soft tissue sarcomas. These findings suggest that radiomic analysis of edema features can provide valuable insights into the prediction of lung metastasis in soft tissue sarcomas.

## 1. Introduction

Soft tissue sarcomas (STSs) encompass a diverse range of malignancies originating from mesenchymal cells. The World Health Organization recognizes more than 50 distinct subtypes within this category. STSs are rare tumors, accounting for approximately 1% of all cancer cases [1]. Despite their low incidence, they pose significant concerns due to their potential for distant metastases, which occur in about 25% of cases and contribute to the majority of deaths; high-grade STSs can exhibit a metastatic rate of up to 50% [2,3,4]. The lungs are the most common site of metastasis, accounting for around 80% of lesions [5].

The prognosis for patients who develop metastases is generally poor. Those who undergo surgical metastasectomy have a 3-year survival rate of less than 50%, while patients who are not eligible for surgery have a survival rate below 20%. The median survival time following the diagnosis of distant metastasis is approximately 11.6 months [2]. The identification of patients with a heightened susceptibility to developing distant metastases holds the potential to enhance the efficacy of therapeutic interventions [6,7].

In a study conducted by White [8], the presence of satellite tumor cells was observed in 10 out of 15 patients with STSs. In 9 out of 15 cases, tumor cells were identified beyond the sarcoma margin within regions exhibiting peritumoral edema and reactive changes as observed on preoperative MRI scans.

By thoroughly investigating the edema, researchers can gain valuable insights into the intricate interactions between the tumor and its surrounding tissue. The edema is closely interconnected with the tumor microenvironment, which encompasses factors such as inflammation, angiogenesis, and tissue remodeling. This comprehensive analysis of the edema can provide additional prognostic information beyond relying solely on the tumor volume. The incorporation of edema analysis in the evaluation of STSs has the potential to aid in patient risk stratification and facilitate personalized treatment decisions.

Despite the significance of the edema in the tumor microenvironment, there is a notable gap in the existing literature. Our literature search on PubMed using the keywords [(“soft tissue sarcoma” OR “soft tissue sarcoma”) AND edema] revealed a lack of studies specifically focused on radiomic features extracted solely from the edema. Therefore, the primary objective of our study is to fill this gap by investigating the potential correlations between radiomic features derived from the edema of STSs and the occurrence of lung metastases.

Through this exploration, our study aims to uncover the prognostic value and clinical significance of these radiomic features in relation to lung metastases in STS patients. By elucidating the role of edema-related radiomic features, we can advance our understanding of STSs and improve patient management strategies. This investigation may also lead to the identification of biomarkers associated with tumor behavior and response. Ultimately, our study seeks to contribute valuable knowledge to the field and enhance the care provided to STS patients.

## 2. Materials and Methods

### 2.1. Dataset

For our study, we employed an open-source anonymized database as the principal data repository (available online: http://doi.org/10.7937/K9/TCIA.2015.7GO2GSKS; accessed on 2 September 2023); this comprehensive dataset consisted of 51 cases of STSs affecting the extremities, which were histologically confirmed [7,9]. Each patient in the dataset had undergone fluoro-D-glucose positron emission tomography and MRI scans as part of their evaluation, conducted between November 2004 and November 2011.

It is important to note that the MRI protocols employed were not standardized across all patients. To ensure consistency in our analysis, we specifically selected T2-weighted fat-saturated (T2FS) or short tau inversion recovery (STIR) MRI scans. The patients were categorized into two groups based on clinical outcomes: “no lung metastases” (group A) and “lung metastases” (group B).

The inclusion criteria required that the selected examinations exhibit distinct segmentations for both the tumor mass and the tumor mass plus the associated edema, while excluding cases where the two segmentations were identical (e.g., cases with no observable edema). In other studies, T2FS and STIR images are deemed comparable in terms of texture analysis; therefore, we grouped them together as a single category [7,10].

Following these criteria, a total of 32 patients were included in our analysis.

### 2.2. Segmentation and Feature Extraction

The segmentations for the examinations were acquired from the aforementioned publicly available database. Each individual segmentation underwent visual evaluation by a radiologist with eight years of experience, and modifications were made as deemed necessary. The 3D Slicer software, version 4.13, was employed for this process [11].

For every exam, the following segmentations were considered:Gross Tumoral Volume (GTV): a segmentation that encompassed only the tumor mass;Edema Tumoral Volume (EDV): this segmentation was derived by subtracting the tumor mass segmentation alone (GTV) from the segmentation that encompasses both the tumor mass (GTV) and the associated edema (see Figure 1).

The extraction of features, i.e., the derivation of features from radiological images, was performed using Pyradiomics 3.0.1 (https://pyradiomics.readthedocs.io; accessed on 2 September 2023), a software library designed for the extraction of radiomic features from medical imaging data [12]. Additionally, a Python script developed by the authors was utilized, ensuring compliance with the Image Biomarker Standardization Initiative (IBSI) standard [13].

The hyperparameters for feature extraction were set with the following values: normalize = True; removeOutliers = 3; binCount = 50; resampledPixelSpacing = 0.8, 0.8, 5.5; interpolator = sitk.sitkBSpline; correctMask = True. All other parameters were kept at their default values. For each examination, the features were extracted individually from each exam in a separate manner.

The radiomic features extracted in this study were categorized into seven main groups: First Order (FOF) Features; Shape Features (SHAPE); Gray Level Co-occurrence Matrix (GLCM) Features; Gray Level Run Length Matrix (GLRLM) Features; Gray Level Size Zone Matrix (GLSZM) Features; Gray Level Dependence Matrix (GLDM) Features; Neighboring Gray Tone Difference Matrix (NGTDM) Features. The definitions and a detailed list of these features can be found in the Pyradiomics feature documentation, available at https://pyradiomics.readthedocs.io (accessed on 2 September 2023).

### 2.3. Feature Selection

The feature selection process aimed to identify and select the most informative features for incorporation into our models. To commence this process, we initiated the identification and elimination of highly correlated features.

This was achieved through the utilization of the Spearman correlation coefficient, where features displaying a correlation value surpassing 0.8 were systematically discarded.

Following this initial step, our approach involved a comprehensive evaluation of potential feature combinations. We commenced this evaluation by considering individual features and then progressively expanding the combination size, ultimately capping it at a maximum of five features. For each combination size, we harnessed the Exhaustive Feature Selection algorithm [14], which meticulously scrutinized all possible combinations; we computed the average area under the receiver operating characteristic curve (AUC) score using a 5-fold cross-validation approach and a Random Forest (RF) classifier. In essence, for each combination size, we identified and selected the combination that yielded the highest average AUC score, thus designating it as the optimal combination for that specific number of features.

Finally, the number and names of the ultimately selected features were determined by identifying the first peak value in the average AUC score (avg_score). This selection process was conducted across the best combinations ranging from 1 to 5 features.

To illustrate with an example, we systematically explored various feature combinations of different sizes to identify the optimal set of features for our analysis:Single Feature Evaluation: When considering single features in isolation, we observed that ‘feature_C’ exhibited the highest AUC of 0.65, outperforming all other individual features.Two-Feature Combinations: Expanding our investigation to pairs of features, we found that the combination of ‘feature_D’ and ‘feature_H’ produced the most favorable result, with an AUC of 0.77. This combination surpassed all other two-feature combinations.Three-Feature Combinations: Continuing our analysis, we explored combinations of three features. Among these, ‘feature_A’ + ‘feature_C’ + ‘feature_F’ yielded the highest AUC of 0.75, demonstrating superior performance when compared to other three-feature combinations.Four-Feature Combinations: Extending our search to combinations of four features, ‘feature_B’ + ‘feature_D’ + ‘feature_F’ + ‘feature_H’ achieved an AUC of 0.71. This particular combination displayed notable predictive power within the set of four-feature combinations.Five-Feature Combinations: Finally, in the context of five-feature combinations, ‘feature_A’ + ‘feature_C’ + ‘feature_F’ + ‘feature_H’ + ‘feature_G’ exhibited the highest AUC of 0.81, outperforming all other five-feature combinations.

After these five steps, we opted for a two-feature combination, ‘feature_D’ + ‘feature_H’, which achieved an AUC of 0.77. This decision was based on the observation that it represented the first peak of the AUC values among the feature combinations, ranging from one to five features.

More details regarding the Exhaustive Feature Selection algorithm and the curves obtained in our analysis are elaborated in the Appendix A (Exhaustive Feature Selection algorithm section, Appendix A).

### 2.4. Modeling and Statistical Analysis

An RF model based on the selected GTV features (RF-GTV) and an RF model based on the selected EDV features (RF-EDV) were compared.

In particular, we performed 100 random subsampling iterations to evaluate the performances of the two models. For each iteration, we randomly split the dataset into training and testing sets; as suggested by Nadeau and Bengio [15], the training set was five times larger than the testing set.

The RF models were trained on the training sets and evaluated on the corresponding testing sets. Performance metrics, such as accuracy, sensitivity, specificity, and AUC, were computed for both algorithms.

The median and interquartile range (IQR) of accuracy, sensitivity, specificity, and AUC were calculated across the 100 iterations for both the RF-GTV and the RF-EDV models.

To compare the two models, we used the Nadeau and Bengio corrected resampled *t*-test for the obtained accuracies. According to [15,16], performing 100 randomized subsampling iterations and the Nadeau and Bengio corrected resampled *t*-test guarantee a close alignment of Type I error with the significance level. Importantly, in contrast to McNemar’s test and the 5 × 2 cross-validation test, this method doesn’t exhibit a heightened Type II error rate. Moreover, when employing a total of 100 runs, the level of replicability reaches a satisfactory threshold, thereby enabling reliable comparisons among diverse algorithms.

The Spearman correlation coefficient was employed to calculate the intercorrelation among the selected features. Additionally, the Mann-Whitney test was utilized to assess statistically significant differences in selected feature values between group A/group B.

The correlation between the clinical features and the clinical outcomes (the “no lung metastases” group and the “lung metastases” group) was subjected to statistical analysis. This analysis employed the Mann-Whitney U test and Fisher’s exact test [17].

To enhance the generalizability of our findings to a wider population, we utilized 10000 stratified bootstrap iterations to calculate 95% confidence intervals (CI) [18], with a particular focus on accuracy and AUC.

The described pipeline was performed using Python version 3.8. For RF, max_depth (the longest path from the root node to the leaf node) was set to 10; the default values were retained for all the remaining parameters.

## 3. Results

### 3.1. Dataset

Our study comprised a cohort of 32 patients, consisting of 14 males and 18 females, with a median age of 60 years (range: 16–83 years). Throughout the follow-up period, 18 patients remained free from lung metastases (group A), while 14 patients experienced lung metastases (group B).

The median duration from the diagnosis to the last follow-up was 684.5 days (range: 377–1329 days) for group A, whereas the median duration from the diagnosis to the onset of metastases or local recurrence was 162 days (range: 29–731 days) for group B.

Regarding histological grades, 18 patients had high-grade sarcomas (8 patients in group A and 10 patients in group B), 13 patients had intermediate-grade sarcomas (10 patients in group A and 3 patients in group B), and 1 patient had a low-grade sarcoma (in group A). Further details on relevant clinical parameters and treatment modalities can be found in Table 1, along with the supplementary information provided in the Appendix A section under “Clinical data”.

The MRI protocols were heterogeneous; T2FS or STIR sequences were used. Additional details regarding the MRI acquisition protocols can be found in the Appendix A section under “MRI data”. Not all individual patients had both STIR and T2FS sequences available. Consequently, we selected the only fluid-sensitive sequence that was accessible for each patient during the analysis [7,10].

### 3.2. Features Extraction

After conducting a visual evaluation, it was determined that the segmentations of 31 exams were suitable for both GTV and EDV; however, in one exam, manual adjustments were made to improve the delineation of the EDV segmentation.

A comprehensive set of 214 radiomics features was extracted, specifically comprising 107 features from the EDV segmentation and 107 features from the GTV segmentation.

### 3.3. Features Selection

In relation to the features extracted from GTV and EDV, after the removal of the highly correlated features, a total of 31 features were retained for GTV and 33 for EDV. Subsequently, the Exhaustive Feature Selection algorithm was iteratively applied, considering the range of one to five features, and identified the first peak in the average AUC score (more details regarding the curves obtained are elaborated in the Appendix A Appendix A). As a result, two features were selected for both GTV and EDV, as shown in Table 2.

### 3.4. Classification Performance

After conducting 100 random subsampling iterations for both the RF-GTV and the RF-EDV models, the resulting performance metrics are shown in Table 3; in particular, the accuracy was 0.83 for the RF-GTV and 0.75 for the RF-EDV.

Based on the results of the Nadeau and Bengio corrected resampled *t*-test, there was no statistically significant difference observed between the accuracies of the two models (*p*-value = 0.433).

Figure 2 presents the ROC curves, along with the corresponding AUC values for both models. These ROC curves and AUC values serve as essential visual and quantitative tools for the assessment of the predictive performance and discriminative capabilities of their respective models, offering valuable insights into their effectiveness in distinguishing between various classes (group A, “no lung metastases” versus group B, “lung metastases”). The ROC curves provide a graphical representation of the models’ trade-offs between sensitivity and specificity across different threshold settings, enabling a nuanced evaluation of their diagnostic or predictive utility. Meanwhile, the AUC values summarize the overall discriminatory power of each model. In particular, the RF-GTV model obtained an AUC of 0.88 and the RF-EDV model achieved an AUC of 0.79.

The bootstrap evaluation, which went through 10000 iterations for calculating the median and 95% CI, revealed the following values:For the RF-GTV: a median accuracy of 0.71 [95% CI: 0.46–0.92], a median AUC of 0.79 [95% CI: 0.50 1.00];For the RF-EDV: a median accuracy of 0.69 [95% CI: 0.43–0.91], a median AUC of 0.73 [95% CI: 0.45 0.94].

Further details regarding the bootstrap results can be found in the Appendix A (Appendix A).

Figure 3 displays the intercorrelation patterns observed among the features selected for both the RF-GTV and the RF-EDV models. These intercorrelations were computed through the application of the Spearman correlation coefficient. The examination of the correlation coefficients revealed values consistently below the value of 0.3, demonstrating a lack of substantial correlation. Such findings emphasize the relative independence of these features within the context of our models.

Figure 4 and Figure 5 illustrate the boxplots representing the selected features in group A and group B; the statistical analysis conducted using the Mann-Whitney test revealed no significant differences among the selected features in terms of the comparison between group A and group B.

## 4. Discussion

This study aimed to compare the predictive ability of radiomic features extracted from the edemas and tumoral masses of STSs in predicting lung metastases. MRI scans of 32 STSs were retrospectively analyzed, of which 18 cases were without lung metastases and 14 cases had lung metastases. A total of 107 radiomic features were extracted from each GTV and EDV segmentation. After feature selection, the feature vectors contained two features for the mass model (original_glcm_Correlation and original_glszm_SmallAreaLowGrayLevelEmphasis) and two features for the edema model (original_firstorder_Kurtosis and original_glszm_SizeZoneNonUniformityNormalized). Random Forest models were created using the selected features, and key performance metrics were calculated. The model based on the mass-related features (RF-GTV) achieved a median accuracy of 0.83 and a median AUC of 0.88, while the model based on the edema-related features (RF-EDV) achieved a median accuracy of 0.75 and a median AUC of 0.79. According to the Nadeau and Bengio corrected *t*-test, the statistical analysis showed no significant difference between the accuracies of the two models.

The Spearman correlation coefficient was used to assess the independence of feature vectors; the results revealed an absence of substantial correlation (Spearman correlation coefficient < 0.3).

In relation to the statistical differences in the distribution of clinical parameters between Group A and Group B, no statistically significant differences were observed for age, gender, grade, and MSKCC type (*p*-value > 0.05). It is noteworthy that, despite references [19,20], which assert that the risk of distant metastases in STSs can range from 20% to nearly 100% based on grading and histological type, our study did not identify any significant correlations between these clinical parameters and the risk of pulmonary metastases. Consequently, the radiomic model demonstrated its capacity to predict outcomes, in contrast to the clinical parameters examined in this study.

In the context of comparing images obtained from the EDV segmentations of individuals without lung metastases (group A) and those with lung metastases (group B), some differences were observed, even though they were not statistically significant. Specifically, Group A exhibited a lower median value of Original FirstOrder Kurtosis, indicating a more peaked distribution of pixel intensities around the mean. Furthermore, Group A exhibited a significantly higher median value of Original GLSZM Size-Zone Non-Uniformity Normalized, indicating the presence of regions within the image with varying sizes or distinct patterns.

In other terms, group A’s images showed a relatively consistent overall texture with localized variations or structures that contribute to the general heterogeneity of the images. In contrast, group B had pronounced variations and irregularities in pixel intensities throughout the region, with regions that exhibited relatively consistent size zones, indicating the presence of distinct histological structures or patterns (e.g., tumor cells arranged in well-defined nests).

To support these findings, the presence of satellite tumor cells within the context of edema and the association of edema and high-grade STSs has been investigated in [8]. Moreover, several previous studies [21,22,23,24,25,26,27] have examined the impact of various factors on MRI (including edema), and have highlighted the prognostic significance of baseline size, heterogeneous signal intensities on pre-treatment conventional MRI sequences, necrotic signals, peritumoral edema and enhancement, and the presence of a tail sign; these studies have also investigated the associations between these features and the histological grading according to the “Fédération National des Centres de Lutte Contre le Cancer” grading system.

Other studies have examined edema [7,26,28,29,30,31,32,33,34,35], but none of these studies have specifically extracted radiomic features solely from the edema region of STSs. In particular, Crombé et al. [26] examined the changes in semantic features before, during, and after neoadjuvant therapy and surgery using MRI. It was found that changes in edema enhancement were associated with the presence of tumor cells beyond the lesion borders, while variations in edema were associated with disease-free survival; however, none of the studied outcomes were associated with the assessment of edema on baseline MRI. Fadli et al. [28] examined the changes in semantic and radiomic features in a cohort of two consecutive pre-therapy MRIs; the findings revealed a significant association between the presence or increase in edemas (assessed semantically) and the occurrence of local recurrence. In another study [30], selected semantic and radiomic features were analyzed in a cohort of patients who underwent two consecutive MRI scans before and after two cycles of neoadjuvant chemotherapy; the analysis revealed that variations in the surrounding edema (measured semantically) were associated with a positive treatment response, defined as a threshold of less than 10% viable cells on surgical specimens.

Our results have been compared with other studies that utilized the same dataset [7,9]. Specifically, Vallières et al. [7], using a model based on four features extracted from FDG-PET/T1 and FDG-PET/T2FS to predict the onset of lung metastases, achieved an AUC of 0.984 using a bootstrap evaluation. Zhao et al. [10], employing a signature based on T2-weighted MRI features to predict the development of metastasis or recurrence, obtained AUC values of 0.8481 and 0.7351, respectively, in the training and validation datasets, using a four-fold cross-validation. Escobar et al. [36] developed a model to predict the onset of lung metastases using MRI sequences, and achieved an AUC of 0.840 using BootstrapOutOfBag. In [37], a methodology utilizing formal logic and radiomics models to predict the risk of metastasis or recurrence yielded an accuracy of 0.74. However, unlike our study, the first three works employed segmentations that included only the GTV, while the fourth study used segmentations that included both the GTV and edema together.

The aforementioned literature has highlighted the scarcity of studies dedicated to the analysis of radiomic features extracted exclusively from edema. In contrast to previous studies, our findings underscore the importance of investigating radiomic features derived solely from edema. This is attributed to the potential they hold in providing valuable insights into the prediction of lung metastases.

The current study has several limitations that need to be acknowledged. Firstly, the sample size used in this study is relatively small, which can reduce the statistical power of the classification outcomes. To mitigate this limitation, we employed 100 random subsampling iterations to assess the performance of the two models. Each iteration involved randomly splitting the dataset into training and testing sets, following the recommendation of Nadeau and Bengio [15], with the training set being five times larger than the testing set. Secondly, the study faced limitations associated with variations in MRI scanning parameters. These differences could potentially introduce batch effects, but they also presented an opportunity to examine the robustness of the methods across diverse image acquisition parameters. Thirdly, we did not employ the DeLong test to compare the AUC values of the models. We made this decision based on concerns raised about the DeLong method, primarily due to its misuse when training and testing the models using the same dataset [38,39]. Instead, we opted to use the Nadeau and Bengio corrected *t*-test to compare the accuracies of the models [16].

In summary, both the RF-GTV and the RF-EDV models exhibited promising potential for predicting the occurrence of lung metastasis in soft tissue sarcomas. Specifically, the model incorporating radiomic features solely extracted from the edema region (RF-EDV) demonstrated the capability to predict lung metastases, although its performance was slightly inferior to the model based on mass-related features (RF-GTV). However, the disparity between the two models did not reach statistical significance.

## 5. Conclusions

These findings suggest that the utilization of radiomic analysis focusing on edema features holds promise in predicting lung metastases in STSs, providing results that are comparable to those obtained from mass-related features. Further investigations involving larger cohorts are warranted to validate the clinical utility of these models.

## Figures and Tables

**Figure 1 diagnostics-13-03134-f001:**
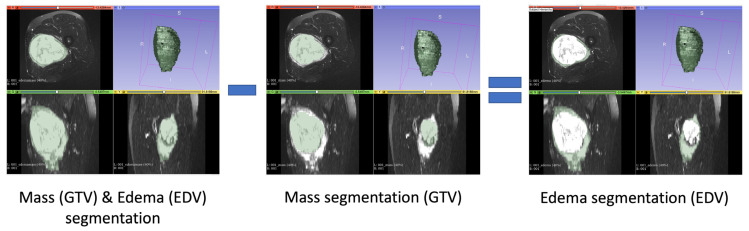
GTV and EDV segmentations; EDV segmentations were obtained by subtracting the GTV segmentation from the segmentation that encompasses both the tumor mass (GTV) and the associated edema.

**Figure 2 diagnostics-13-03134-f002:**
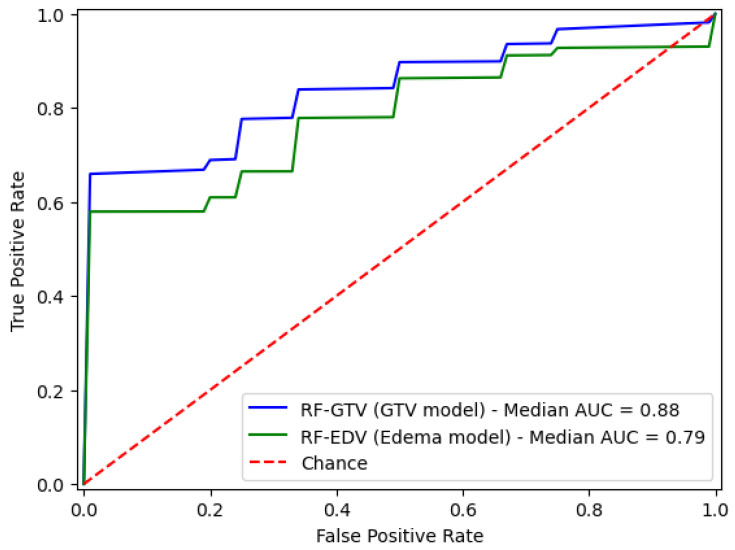
ROCs and AUCs for the RF-GTV (GTV model) and RF-EDV (edema model).

**Figure 3 diagnostics-13-03134-f003:**
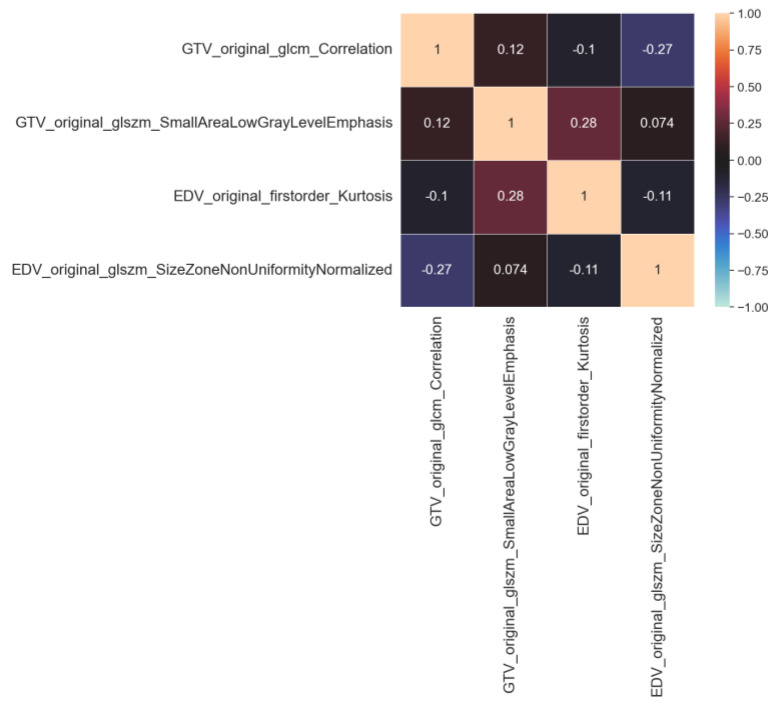
Intercorrelation among the selected features for both Gross Tumor Volume (GTV) and Edema Tumor Volume (EDV).

**Figure 4 diagnostics-13-03134-f004:**
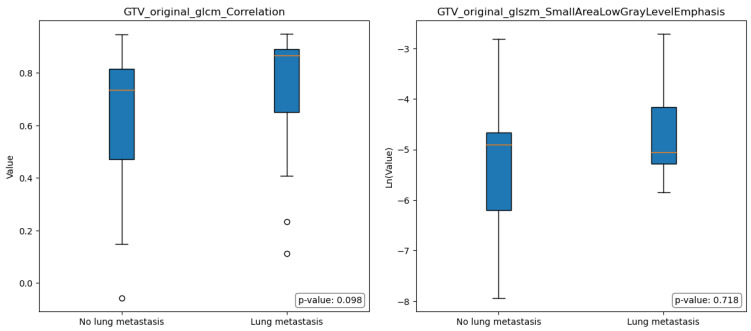
Boxplots for selected features for Gross Tumor Volume (GTV) model; Mann-Whitney *p*-value for statistically significant differences of value distribution. The horizontal orange line represents the median.

**Figure 5 diagnostics-13-03134-f005:**
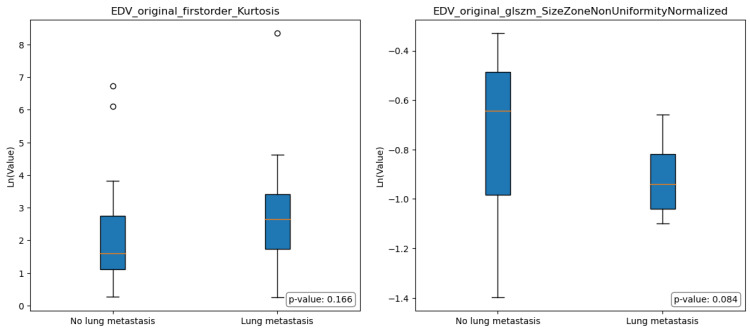
Boxplots for selected features for Edema Tumor Volume (EDV) model; Mann-Whitney *p*-value for statistically significant differences of value distribution. The horizontal orange line represents the median.

**Table 1 diagnostics-13-03134-t001:** Clinical parameters. (* *p*-value for statistically significant differences of value distribution in Group A and Group B; age—Mann–Whitney; gender ratio, grade radio and MSKCC type—Fisher’s exact test).

	Group A (No Lung Metastases)	Group B (Lung Metastases)	*p*-Value *
Number of patients	18	14	-
Gender ratio (M/F)	5/13	9/5	0.072
Age, y, median (range)	53.5 (16–83)	62.5 (44–74)	0.106
Grade ratio (Low/Intermediate/High)	1/9/8	0/4/10	0.216
MSKCC type (Fibrosarcoma/Leiomyosarcoma/Liposarcoma/MFH/Synovial sarcoma/Other)	1/6/3/3/3/2	0/3/2/8/1/0	0.238

**Table 2 diagnostics-13-03134-t002:** Selected features for Gross Tumor Volume (GTV) and Edema Tumor Volume (EDV).

Selected Features	
Gross Tumor Volume (GTV)	Edema Tumor Volume (EDV)
original_glcm_Correlation	original_firstorder_Kurtosis
original_glszm_SmallAreaLowGrayLevelEmphasis	original_glszm_SizeZoneNonUniformityNormalized

**Table 3 diagnostics-13-03134-t003:** Classification performance on 100 random subsampling iterations.

	RF-GTV Median [Interquartile Range]	RF-EDV Median [Interquartile Range]
Accuracy	0.83 [0.17]	0.75 [0.17]
Sensitivity	0.67 [0.50]	0.67 [0.50]
Specificity	1.00 [0.33]	0.80 [0.33]
AUC	0.88 [0.23]	0.79 [0.38]

## Data Availability

The dataset, the code and the segmentations used in this article can be provided upon contact with the corresponding author.

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
