# Peer review of "The Impact of Edema on MRI Radiomics for the Prediction of Lung Metastasis in Soft Tissue Sarcoma"

_diagnostics, 2023, doi:10.3390/diagnostics13193134_

Round 1

Reviewer 1 Report

Thank you for authors for this interesting paper.

However, some questions remain to be answered:

-The data set is quite old. Has the tecnhology/quality in the MRI scanning (how many Teslas, quality of the imaging results, accuracy, etc.) changed  after that? So are the results valid anymore? Why were so old date set chosen? Most of the sarcoma patients metatasize before two years of the diagnosis, so more recent dataset would have given as accurate survival rates as old ones. 

-The paragraph "Feature selection" is unclear to reader who is not 100% familiar with the process. Does it describe the feature group selection or single feature selection? One feature group was not chosen to this analysis, so it should be clarified in the text.

-The grade was quite different between the study groups according to the table 1. How much of the results can be explained with this issue? Do you know the histology and other clinical factors of the tumors (those should also be reported as explanatory factors)? When concerning these clinical factors can the radiomics offer something more to a clinician?

-The titles Feature selection and Features extraction and Selection are very similar. Please clarify those better to be different issues.

-institutional review board statement is missing and should be mentioned.

-Author contributions and conflicts of interests are missing and should be mentioned.

-line 163: please correct 1 patient (not patients).

-Line 197: please remove : It is not needed in this context. 

Author Response

I would like to express my gratitude for your inquiries and suggestions. The MRI sequences and techniques employed in our study align with clinical practices, within real-world contexts. Supplementary details pertaining to the specific MRI sequences and techniques utilized throughout our research are provided within the supplementary table.

Regarding the MRI magnetic field, it is noteworthy to reference the article at https://doi.org/10.1088/1361-6560/aa8a49, which incorporated some exams from the same dataset. This article reported that the images were acquired using a scanner featuring a 1.5 Tesla (T) magnet.

Our selection of this dataset stems from its open-source nature and public accessibility. This deliberate choice enabled us to conduct comparisons between our outcomes and those of fellow researchers who also engaged with the same dataset. Such comparisons serve to fortify our conclusions, particularly in relation to the gross tumor volume model. We have added these comparative analyses within the discussion section of our paper.

The paragraph titled "Feature selection" describes the process of selecting the best combination of individual features, which were analyzed one by one and added to the selected feature group. We acknowledge that this aspect was not sufficiently clarified in the original text, leading to confusion.

In response to your comment, we have revised the text in the article to provide a clearer explanation of the feature selection process, with an example. Additionally, we have included a supplementary materials section that elaborates on this process in greater detail.

Thank you for your inquiry. Concerning the risk of distant metastasis, as indicated in ISBN-13: 978-92-832-2413-6 and DOI: 10.1111/j.1365-2559.2005.02284.x, this risk can vary significantly, ranging from 20% to 100%, depending on tumor grading and histological type. However, in our study, we did not identify statistically significant correlations between grading or MSKCC type and the risk of metastasis. We systematically examined various clinical factors, including grading and MSKCC type, to explore potential statistically significant relationships between these clinical parameters and the outcome (the onset of pulmonary metastases) using Mann-Whitney U test and Fisher's exact test (as suggested by DOI: 10.5395/rde.2017.42.2.152). Our findings revealed that these clinical parameters were not correlated with the outcome. Consequently, the radiomic model demonstrated the ability to predict the outcome, in contrast to the clinical parameters analyzed in this study.

We have included the results of these statistical tests and above mentioned references in the text.

- Thank you for this suggestion. We have taken measures to better clarify and differentiate between the titles and the meaning of Feature extraction and Features selection in the article.

- For this particular study, an institutional review board statement was not required, as the research involved a series of examinations extracted from an open-source archive and evaluated retrospectively. However, it is noteworthy that ethical approval was sought for the usage of the open-source archive, as described in DOI: 10.1088/0031-9155/60/14/5471, where it states, 'Subsequent to research ethics board (REB) approval, a database of 51 patients with histologically proven primary STSs of the extremities was retrospectively retrieved.' This ensures adherence to ethical standards in data usage for our research.

- We have taken the necessary steps to include author contributions and conflicts of interest in the manuscript.

Comments on the Quality of English Language

-" line 163 : please correct 1 patient (not patients). "

Thanks for the suggestion. We corrected the typo.

- " Line 197 : please remove : It is not needed in this context. "

We have fixed the error.

Reviewer 2 Report

Authors have investigated an important area of predicting Lung Metastasis in Soft Tissue Sarcoma (STS)- a type of tumor in lungs. Whereas the statistical features and result analysis methods used are appreciable, but the methodology used for prediction seems irrelevant and ambiguous. Authors want to emphasize the impact of edema on STS; but, as discussed Segmentation and Feature Extraction part of the paper edema tumoral volume (EDV) is derived by subtracting gross tumoral volume (GTV) from mass & edema volume. Thus, the features obtained for EDV have to align with GTV. Had the edema volume been obtained by altogether a different apporaoch than a mere subtraction, then its impact could have been solely estimated. However, that is not the case here,

It is stated that an Exhaustive Feature Selection algorithm was used to select two features out of 107 features for GTV and EDV. Which is this algorithm and why only original_glcm_Correlation & original_glszm_SmallAreaLowGrayLevelEmphasis for GTV and Kurtosis & Non-uniformity for EDV have been selected? It should be justified.

The significance of figure 2 and figure 3 should be comprehensively explained. 

English language of the manuscript may be checked once again to remove some flaws here and there. For example, on line 131 "A RF model based on..." should be replaced by "An RF model..."

Author Response

- We appreciate your valuable feedback. In response to your suggestion, we have taken steps to provide a clearer explanation within the text regarding the procedure for obtaining segmentations that exclusively contain edema (EDV). Specifically, it's important to note that the segmentations for edema do not include any voxel content representing the gross tumoral volume (GTV). Therefore, the features extracted from GTV [including First Order (FOF) Features, Gray Level Co-occurrence Matrix (GLCM) Features, Gray Level Run Length Matrix (GLRLM) Features, Gray Level Size Zone Matrix (GLSZM) Features, Gray Level Dependence Matrix (GLDM) Features, Neighbouring Gray Tone Difference Matrix (NGTDM) Features] are entirely independent of the features extracted from EDV.

To further illustrate this independence, we have conducted an analysis to assess the correlation between the features selected from edema segmentations and those selected from gross tumoral volume segmentations. As indicated in Figure 3, the Spearman correlation coefficient is consistently below 0.29, demonstrating a lack of substantial correlation.

We trust that this clarification addresses your concerns regarding the potential overlap of features between EDV and GTV, and underscores the distinctiveness of the feature sets derived from each.

- We have revised the text to provide a clearer description of our feature selection process, which led to the selection of two features for GTV and two for EDV. Additionally, we have included an explanation of the Exhaustive Feature Selection algorithm in the supplementary material to provide a more comprehensive understanding of its functioning.

- We have taken steps to provide a more comprehensive explanation of the significance of Figures 2 and 3 in the text.

Comments on the Quality of English Language

"English language of the manuscript may be checked once again to remove some flaws here and there. For example, on line 131 « A RF model based on… » should be replaced by « An RF model… » "

Thanks for this suggestion. We have addressed the identified errors and typos as suggested.

Reviewer 3 Report

The authors evaluate whether radiomic features extracted solely from the edema of soft tissue sarcomas could predict the occurrence of lung metastasis and also they compare the predictive ability of radiomic features extracted from the edema and tumoral mass of soft tissue sarcomas in predicting lung metastasis. The topic of the work is very good and the result of the research can be very effective in predicting lung metastasis but there are some issues in the manuscript which must be revised as follows:

1 - Why is the number of patient samples selected low? Is it possible to be satisfied with this number and how does it confirm the result?

2 - What is the importance of this research compared to previous works? What does it add to previous research? Please explain in the text.

3 - In the lines 132 and 187, it is mentioned that 100 random samplings were done. I am concerned about the choice. It would be better to choose a higher sampling. What is the reason for choosing this number?

4 - The reason for selecting this dataset in line 68 is not clear. The results are from 2004 to 2011. Why was this interval chosen? Why is the most recent data not selected? The average age of patients is also very high. It seems a good sampling was not chosen. Please explain in the text why this was done?

5 - Please explain about the algorithm used in the work, as well as the duration of the work and the hardware used. How is the sampling error calculated and what order is it?

 6- Figure 1 is not clear. It is better to provide a clearer form or to present a form in a different way. Although it seems that instead of this figure, the explanation in the text is also sufficient.

7 - Aren't the error bars of Figures 4 and 5 too high? Is it because of sampling numbers? please check this.

Author Response

1) Unfortunately, the rarity of the pathology under investigation constrains the availability of large patient cohorts. Indeed, many published radiomics studies concerning STSs feature sample sizes of approximately 30 patients or even fewer (e.g., 10.1088/1361-6560/aa8a49, 10.1002/jmri.25791, 10.1016/j.acra.2018.09.025, 10.2478/raon-2019-0041, 10.1002/jmri.26753, 10.1088/1361-6560/ab9e58, 10.3389/fonc.2022.879553). However, to facilitate the generalizability of our findings to the broader population, we added and employed the calculation of 95% confidence intervals through 10,000 bootstrap iterations, particularly for accuracies and AUCs. These confidence intervals allow us to estimate the results for the entire population based on a limited sample size, enhancing the robustness of our conclusions. Further details for the bootstrap results and related histograms are depicted in Supplementary materials.

2) Thanks for the feedback. In response to this point, we have incorporated a comparison with other relevant studies in the field of radiomics pertaining to STSs in the discussion section (also other studies utilizing the same dataset). This comparative analysis elucidates the unique contributions of our work and underscores the added value it brings to the existing body of research.

3) We selected the number of random samplings, specifically 100, in accordance with the methodology proposed by Nadeau and Bengio (DOI: 10.1007/978-3-540-24775-3_3), as detailed in their work. However, as previously mentioned, we have also added an assessment involving 10000 bootstrapping iterations to further substantiate our findings.

4) We selected this dataset due to its open-source nature and public accessibility. This deliberate choice facilitated the comparison of our results with those of fellow researchers who also utilized the same dataset, enhancing the robustness of our conclusions, particularly with regard to the gross tumor volume model. We have incorporated these comparative analyses in the discussion section of our paper to provide comprehensive insights. Moreover, the MRI sequences and techniques employed in the dataset align with clinical practices, within real-world contexts.

Regarding the age of patients, it is important to note that the inclusion criteria for the dataset were outlined in the text and were independent of age factors. Nevertheless, as detailed in DOI: 10.1002/pbc.23252, it is worth highlighting that the average age for STSs tends to be relatively high.

5) Details concerning the algorithm employed for feature selection, hardware specifications, and the duration of our work can be found in the Supplementary Materials.

Regarding the calculation of the sampling error, we chose to utilize the 95% confidence interval computed through the bootstrap methodology. This approach was chosen because estimating the sampling error would have required access to the entire population's standard deviation, which was not obtainable.

Utilizing the 95% confidence interval via bootstrap methodology offers several advantages in the fields of machine learning and radiomics compared to traditional sampling error calculations. Firstly, it provides a more robust estimation of uncertainty, especially when dealing with limited sample sizes. Additionally, it accommodates non-normality in data distributions, which is common in radiomic datasets. Lastly, it facilitates the evaluation of complex models and feature sets, enabling more reliable generalization from limited datasets.

6) Following the suggestion, we have revised Figure 1 to enhance its clarity and presentation. Additionally, we have also made corresponding improvements to the explanatory text to ensure that the concepts are more comprehensible.

7) The box plots in Figures 4 and 5 display broad and overlapping intervals concerning both minimum and maximum values, as well as interquartile ranges, for features pertaining to both the 'no metastasis development' group and the 'metastasis risk' group. This is further indicated by the lack of statistical significance, as confirmed by the Mann-Whitney p-value. Such data demonstrate that individual features, taken in isolation, are not predictive of the assessment of the risk of pulmonary metastasis. However, when incorporated into a classification model, such as Random Forest, they enable the prediction of metastasis risk.

Round 2

Reviewer 1 Report

Thank you for the corrections. Nothing more to clarify.

Reviewer 2 Report

Authors have addressed the suggested changes and incorporated them in the revised manuscript.

Reviewer 3 Report

While thanking the authors, the manuscript in its current form is suitable for publication.